# Male Predominance of Gastric Cancer among Patients with Hypothyroidism from a Defined Geographic Area

**DOI:** 10.3390/jcm9010135

**Published:** 2020-01-03

**Authors:** Maria Pina Dore, Alessandra Manca, Maria Carolina Alfonso Pensamiento, Alessandro Palmerio Delitala, Giuseppe Fanciulli, Andrea Fausto Piana, Giovanni Mario Pes

**Affiliations:** 1Dipartimento di Scienze Mediche, Chirurgiche e Sperimentali, University of Sassari, 07100 Sassari, Italy; alessandra.manca@aousassari.it (A.M.); aledelitala@tiscali.it (A.P.D.); gfanciu@uniss.it (G.F.); piana@uniss.it (A.F.P.); or; 2Baylor College of Medicine, One Baylor Plaza, Houston, TX 77030, USA; 3Facultad de Ciencias Médicas, Universidad de San Carlos de Guatemala, Guatemala 01012, Guatemala; carolinealpen28@gmail.com; 4Endocrine Unit, Azienda Ospedaliero-Universitaria, 07100 Sassari, Italy; 5Sardinia Longevity Blue Zone Observatory, 08040 Ogliastra, Italy

**Keywords:** gastric cancer, hypothyroidism, *Helicobacter pylori* infection

## Abstract

In the past, hypothyroidism has been associated with an increased susceptibility to gastric cancer (GC). Although several epidemiological studies have corroborated this association, a precise mechanistic explanation remains elusive. In this study, this hypothesis was tested by using a large database of subjects who underwent upper endoscopy for various reasons. This was a retrospective, case-control, single-center study. Subjects with GC (cases) were compared with subjects without (controls), according to hypothyroidism status. Overall, the prevalence of GC was 0.73% in the total cohort and was significantly higher in males compared to females (1.4% versus 0.4%, *p* < 0.0001). Multivariate logistic regression analysis confirmed an increased risk in males with hypothyroidism (OR 5.10; *p* < 0.0001) after adjusting for potential confounders, especially *H. pylori* infection. Interestingly, only hypothyroidism and not treatment with levothyroxine was a significant predictor of GC, ruling out a possible direct carcinogenic effect of the replacement therapy. The present study suggests a male-restricted association of gastric carcinogenesis with a hypothyroid state. If the results of this study are confirmed by longitudinal studies, an attractive perspective could open up for the better management of males with concomitant hypothyroidism and a higher risk of GC.

## 1. Introduction

Gastric cancer (GC) is the third most common cause of cancer among males and the fifth among female worldwide with 950,000 new cases every year [1,2]. Although the incidence of GC shows a considerable geographic variation, its distribution may vary, even within the same geographical area [3]. For example, it has been reported that early exposure to environmental risk factors in a given population is capable of influencing mortality and incidence rates of GC to a greater extent than the genetic background [4]. Despite the overall decline of GC incidence, an unexpected rise in the number of new cases has been registered among generations born after the fourth decade of the 20th century [5].

In Italy, the *Associazione Italiana di Oncologia Medica* (AIOM) has estimated a total of 12,700 new cases of GC in 2018 with remarkable regional variations [6]. According to an epidemiological report published in 2014, the distribution pattern of GC in Northern Sardinia is similar to that reported in other Western populations with a stable incidence rate in both sexes [7].

The events leading to gastric carcinogenesis are mostly triggered by *Helicobacter pylori* infection, autoimmune atrophic gastritis, salted/salty and smoked foods, smoking, occupational exposure to dusts, nitrogen oxides, N‒nitroso compounds, ionizing radiation, and asbestos, among others [8,9,10,11,12,13].

It has been postulated that iodine may also play a role in the carcinogenesis of the stomach [14]. For example, in the past it was observed that goiter associated with iodine deficiency was a risk factor for the occurrence of GC in Switzerland [15]. Hypothyroidism shows a variable geographical distribution, depending on the specific area and the population selected, and is greater where the concentration of iodine in the soil is low. These areas, like those where autoimmune thyroid disease is frequent, are often characterized by a high incidence of GC, suggesting a non‒random association of the two conditions [16]. In addition, in a cohort of 29,584 Chinese adults, including 12,970 males, patients with goiter showed a statistically significant association with GC after adjusting for age, sex, regular tobacco smoking, body mass index (BMI) and family history of stomach cancer [17]. On the other hand, the frequency of thyroid disorders was found to be increased in patients with GC compared with healthy controls in a study conducted in Turkey [18]. Similar findings were also obtained by Syrigos et al., who found a positivity for antithyroid antibodies more frequently in patients with GC compared with control subjects [19]. Conversely, in patients with autoimmune thyroid disease a higher incidence of atrophic body gastritis, a well-known risk factor for GC was detected [20]. A study in animal models demonstrated that the administration of thyroxine during the development of disease activity was able to induce a major reduction of anti‒parietal cell antibodies and the incidence of gastritis [21]. Moreover, it has been reported that GC accumulates triiodothyronine (T3) through the overexpression of transthyretin involved in cellular T3 import [22]. The increased intracellular T3 concentration directly contributes to cancer progression, by different pathways [23]. In human surveys, similarly, iodine prophylaxis was effective in decreasing the incidence and death rate for GC in iodine-deficient areas [24]. In a cohort study of 163,972 Danish patients with a diagnosis of either hyperthyroidism or hypothyroidism, an elevated risk of cancers, mostly gastrointestinal, was observed compared with the general population [25]. Moreover, in Italy, a higher prevalence of GC was reported in farmers and in populations living in mountainous and hilly areas with an iodine‒restricted diet with respect to fishermen [26]. Another report showed that iodine deficiency impairs immunity [27] and it might reduce the endogenous immunological defense against tumor cells.

In this study we aimed to evaluate the relationship between GC and thyroid disorders according to sex and *H. pylori* infection.

## 2. Experimental Section

This was a retrospective case‒control single‒center study. Records of adult patients undergoing upper endoscopy for any reason, from January 2002 to December 2018, at the Gastroenterology Section, Department of Internal Medicine, University of Sassari, Italy, were retrieved for the analysis. A trained gastroenterologist completed the charts before the procedure. Data included demographic information, anthropometric measures, marital status, occupation, and smoking habits. In addition, a complete clinical history, all medications taken, and previous diagnoses of any morbidity were collected. The exclusion criteria considered were incomplete data of *H. pylori* status, gastric histology, ongoing medications, and associated disorders.

The study was conducted in accordance with the Declaration of Helsinki, and the protocol was approved by the Local Ethics Committee “*Comitato Etico ASL n° 1 di Sassari*” (Prot N° 2099/CE).

### 2.1. Helicobacter pylori Infection and Histology

Gastric mucosa samples from the antrum, angulus, and the corpus of the stomach were obtained from each patient. Based on macroscopic appearance, additional biopsy specimens were taken from any suspected lesion. Biopsy specimens for histology were fixed immediately in 10% buffered formalin and stained with hematoxylin–eosin and Giemsa stains [28]. Specimens were analyzed throughout the study period by the same gastrointestinal pathologist. The presence of the bacteria in at least one biopsy confirmed the *H. pylori* status. According to the histology features, gastritis was labeled as active, chronic, and follicular with or without intestinal metaplasia, dysplasia, atrophy, and/or additional characteristics [29]. In the case of chronic active gastritis and absence of *H. pylori* in the specimens, the infection was assessed by additional non-invasive tests such as the ^13^C‒urea breath test or by the *H. pylori* antigen stool test.

### 2.2. Thyroid Disorders

Diseases related to the thyroid were classified as hypothyroidism and hyperthyroidism. In addition, the presence of goiter and/or nodules in the gland as well as previous thyroidectomy was recorded. Patients with transient thyroid disorders were not considered for the analysis. Treatment with levothyroxine or methimazole was matched with thyroid disorders in order to minimize bias.

### 2.3. Statistical Analysis

Hashimoto’s thyroiditis associated with low levels of thyroid hormones was included in the hypothyroidism category and analyzed according to the presence of GC. The presence in gastric specimens of intestinal metaplasia and/or follicular gastritis, and/or dysplasia, and/or atrophy associated with activity, and *H. pylori* was considered a long-lasting infection. The presence of intestinal metaplasia, and/or follicular gastritis, and/or dysplasia, and/or atrophy in the absence of activity was considered as a stigma of a previous *H. pylori* infection. For the purpose of this study, only adenocarcinomas were considered. Additional variables included in the analysis were: sex, age, and BMI calculated as weight/height² (kg/m²), and categorized as underweight (<18.0 kg/m²), normal (between 18.5–24.9 kg/m²), overweight (between 25.0–29.9 kg/m²), and obesity (≥30.0 kg/m²). The area of residence of study participants was categorized as urban or rural; in addition, residences were also clustered according to historic geographic regions of Sardinia. Occupation was subdivided, as previously described, as professionals (Class I); technical and administrators (Class II), clerks and sales technicians (Class III); semiskilled and unskilled workers and uneducated farmers and shepherds (Class IV); housekeeping, students, and retired (Class V) [28]. Marital status comprised single, married, widowed, and divorced patients. According to smoking habits, patients were classified as (i) never and (ii) former/current smokers. Multivariate logistic regression models were used to estimate the risk of GC under the assumption of hypothyroidism as a predictor and adjusting for potential confounders. Odds ratios (ORs) and their 95% confidence intervals (CIs) were calculated by exponentiating the regression coefficients. All statistical analyses were performed using SPSS statistical software (version 16.0, Chicago, IL, USA). *p*-Values lower than 0.05 were considered statistically significant.

## 3. Results

### 3.1. Study Participants

Characteristics of the study population are shown in Table 1. A total of 8709 charts were available for the analysis including 5622 female (65%) and 3087 male (35%) patients, respectively. There were 547 (6.3%) patients affected by hypothyroidism with the majority being female (89.9%; *p* < 0.0001) (Table 1). The pattern of levothyroxine use across age decades is represented in Figure 1 for male and female participants. As expected, replacement therapy was more common in females. Differences between male and female patients were statistically significant for age, marital status, BMI, smoking habits, occupation, and geographic area (Table 1).

*Helicobacter pylori* was detected in gastric biopsies of 4454 (51.1%) patients and its prevalence was equal in males and females as well as the detection of stigmata of a previous infection. Among them 2370 patients displayed intestinal metaplasia and or dysplasia and or atrophy. In the remaining patients with an active‒chronic gastritis *H. pylori* was not detected in 125 and the infection was confirmed by ^13^C-UBT or stool antigen assay (Appendix A). Overall, gastric adenocarcinoma was found during the endoscopic procedure in 64 (0.73%) patients and, as expected, the prevalence was statistically higher in males compared with females (1.4% versus 0.4%, *p* < 0.0001). In addition, a total and a partial gastrectomy for a previous diagnosis of GC was observed in 19 and in 17 patients, respectively (Table 1).

In Table 2, the thyroid disorders affecting patients included in the study are listed. The majority were represented by hypothyroidism (1.8% in males and 8.7% in females) and thyroidectomy (0.9% in males and 0.4% in females), whereas there were very few cases with goiter (13 cases), nodules in the gland (10 cases), and hyperthyroidism (19 cases). Interestingly, in males GC was more common in patients with hypothyroidism compared to patients without (9.5% versus 1.7%). There was no one case of hyperthyroidism associated with GC. On the contrary, a high prevalence (92.2%) of adenocarcinomas detected during endoscopy were positive for *H. pylori* infection and/or for premalignant lesions such as metaplasia and/or dysplasia and/or atrophy. The proportion of patients exposed to levothyroxine treatment was similar in patients with and without GC.

The distribution of *H. pylori* infection was higher in rural areas of Northern Sardinia, as shown in Figure 2. As expected, the highest prevalence of hypothyroidism was detected in the Sardinian historical regions of Barbagia and Baronia, located in the mountain area (Figure 3).

### 3.2. Risk Factors for Gastric Cancer

Multivariate logistic regression analysis allowed further assessment of the association between GC and its predictors by adjusting for all covariates. The risk associated with GC was significantly higher in males (OR, 5.105; 95%CI 1.688–15.44) compared with females (OR, 0.349; 95%CI 0.047–2.594) with hypothyroidism. Age appeared to be a significant risk for GC in both sexes (Table 3). A considerable risk was also observed in patients with a previous and/or long‒lasting *H. pylori* infection (presence of metaplasia and/or dysplasia and/or atrophy in the gastric mucosa), higher in males (OR, 2.406, 95%CI 1.456–3.978) compared with females (OR, 1.845; 95%CI 0.826–4.121).

## 4. Discussion

The hypothesis that a lower thyroid function should be listed among the factors associated with GC was advanced more than two decades ago [14]. Several mechanisms have been proposed to account for this association, mostly hinged on iodine deficiency and the ensuing reduction of antioxidant defense [30], as well as the increased absorption of nitrates and related compounds, which are known to promote gastric tumorigenesis [31], and impaired immunity which may alter the endogenous immunological defense against tumor cells [27]. Over the years, a number of epidemiological studies have explored the plausibility of the association between an underactive thyroid and GC, yielding variable results. A study by Goldman et al. which recruited only females, showed a standardized mortality ratio for GC of 5.8 (95%CI 1.2–17.0) in females with Hashimoto thyroiditis and 1.8 (95%CI 0.5–4.7) in those with idiopathic hypothyroidism [32]. In the Chinese study, goiter was associated with a marginal increase in the risk of gastric non-cardia adenocarcinoma [17]. In a small sample of Iranian patients with gastric adenocarcinoma, a higher prevalence of goiter and lower circulating levels of T3, although not T4, compared with controls, were reported [33]. In the Danish study of Kirkegård et al. the overall risk of GC was increased compared with the general population (standardized incidence ratio 1.49; 95%CI 1.26–1.75) [25]. This association was even stronger in the first year following the diagnosis of hypothyroidism (SIR 3.12; 2.26–4.21). Other authors reported a greater frequency of antithyroid antibody titers in patients with GC, although these immune markers do not always indicate a functionally relevant thyroid disease [19]. On the other hand, a study conducted in Northern Italy was unable to detect a significant relationship between thyroid disorders, including goiter, and the occurrence of stomach cancer [34]. However, the study power was inadequate given the small number of cases and controls with available thyroid status, making a type II error almost inevitable.

In the present study, after adjusting for age, sex, residence area, marital status, occupation, BMI, tobacco smoking, previous and current *H. pylori* infection, we found that a clinical history of hypothyroidism was associated with an increased risk (OR: 5.105; 95%CI 1.688–15.44) of GC in males, whereas the estimate was distributed around the null in females, indicating no excess GC risk. Although the CI around the OR for men is wide, the risk was nearly six-fold increased in hypothyroid men compared to euthyroid men suggesting a non-random association. Since, in general, hypothyroidism is more common in females, one may ask why, in our study, the observed increase in cancer risk was significant only among males. A possible interference by some common confounders was excluded by the multivariate analysis: for instance, although males were significantly older than females, this difference was not of such an extent as to justify the large gap observed in cancer risk. One possible explanation may reside in a shorter exposure to hypothyroidism before the replacement therapy in females compared with males. It is a common notion that females tend to see doctors more often than males [35]. Especially in Sardinia, given the high prevalence of thyroid disorders [36], family physicians prescribe serum thyroid function tests routinely. Detection of abnormal tests will prompt levothyroxine treatment. Instead, it is reasonable that males, owing to their reluctance to see doctors, may have a delay in the replacement therapy.

Confounding due to *H. pylori* infection can hardly explain the higher risk of GC among males with hypothyroidism, as the frequency of a previous *H. pylori* infection, as well as the detection rate of this micro-organism in gastric biopsies, was nearly identical in both sexes. The significant greater proportion of tobacco smokers among males, observed in the univariate analysis disappeared in the multivariate analysis. The adjustment for overweight/obesity did not meaningfully change the observed association between hypothyroidism and GC in males. Finally, a direct action of levothyroxine was ruled out by the multivariate analysis.

Under the assumption of a potential cancer risk being equal in both sexes upon exposure to hypothyroidism, the most reasonable hypothesis is that females exposed to hypothyroidism are less likely to develop GC than males, irrespective of *H. pylori* infection and other environmental factors, due to some protection by sex hormones. Studies on animal models as well as in humans indicate that estrogen may protect against GC [37,38] and this mechanism may also be operative in perimenopausal females who, in addition to levothyroxine, may resort to replacement hormone therapies to delay menopause. This notion is consistent with the results of clinical trials showing an increased incidence of GC in females treated with tamoxifen, a known anti-estrogen [39]. However, this information was not available in our study.

Our study has several strengths and limitations. First, the study was performed using a large database and thyroid status was not merely self‒reported but assessed by endocrinologists, thus minimizing the possibility of misclassification. On the other hand, some patients with subclinical hypothyroidism may have escaped proper classification. In addition, the lack of follow-ups prevented us analyzing the influence of hypothyroidism duration before the onset of GC. Finally, the unavailability of dietary data made it impossible to assess the influence of this major risk factor on gastric carcinogenesis. In particular, sex-specific differences in food intake (e.g., less alcohol and more fruit and vegetables among females) could explain the relative protection of females from developing gastric malignancy. Unfortunately, dietary habits were not recorded in our database, therefore we were unable to control for this variable, which remains a potential unmeasured confounder. On the other hand, it cannot be completely ruled out that the increased risk of gastric carcinogenesis observed in males with hypothyroidism was due to mere chance, owing to the small number of cancers recorded, although the effect size revealed by the multivariate analysis was large enough to indicate the presence of a non-random phenomenon.

## 5. Conclusions

This study shows that a hypothyroid status is significantly associated with GC in males. If these findings are confirmed by larger, longitudinal studies, an attractive perspective could open up: a prompt diagnosis and medical treatment of thyroid disease would not only result in escaping hormonal failure, but also in reducing the risk of GC. The subgroup of males at risk of this malignancy with a concomitant hypothyroidism would benefit from health measures to reduce the impact of this serious malignancy.

## Figures and Tables

**Figure 1 jcm-09-00135-f001:**
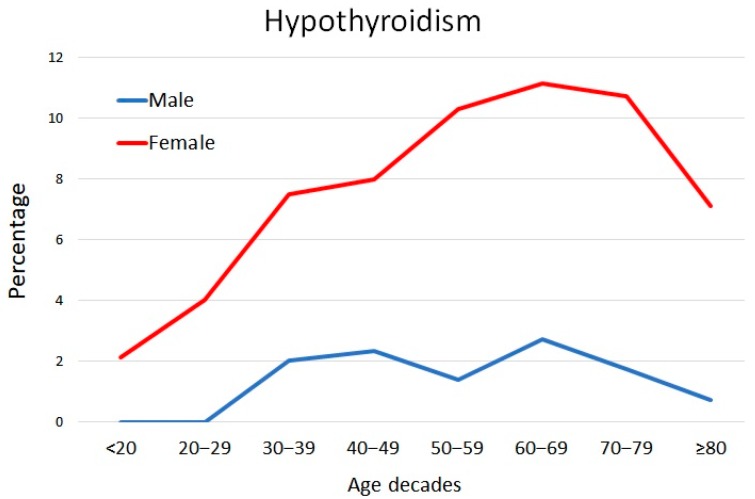
Levothyroxine use by age decades in males and females participating in this study.

**Figure 2 jcm-09-00135-f002:**
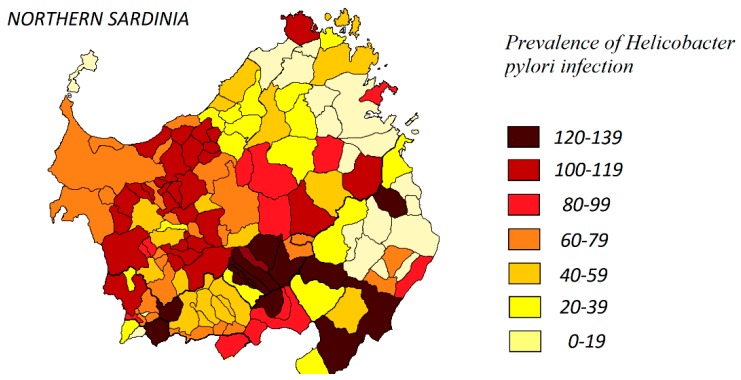
Distribution of *H. pylori* infection in Northern Sardinia according to the study findings.

**Figure 3 jcm-09-00135-f003:**
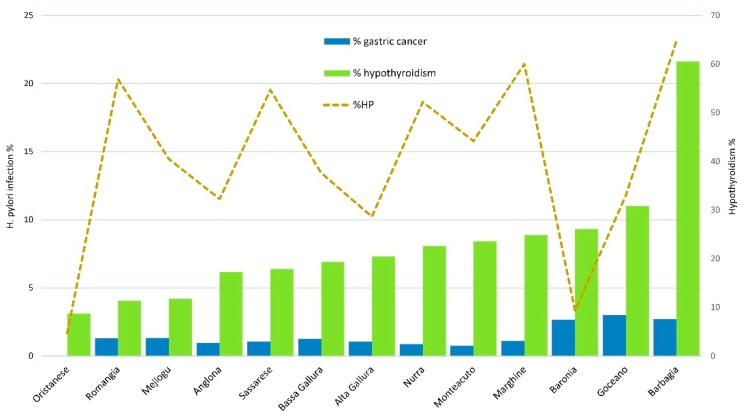
Distribution of hypothyroidism, gastric cancer, and *H. pylori* infection across Sardinian historical subregions.

**Table 1 jcm-09-00135-t001:** Characteristics of the 8709 study participants.

Variables	Males	Females	*p*-Value
*Total no. of patients*	3087	5622	‒
*Age decades*			<0.0001
<20	59 (1.9%)	140 (2.5%)	
20–29	251 (8.1%)	620 (11.0%)	
30–39	445 (14.4%)	785 (14.0%)	
40–49	514 (16.7%)	975 (17.3%)	
50–59	567 (18.4%)	1056 (18.8%)	
60–69	659 (21.3%)	1112 (19.8%)	
70–79	457 (14.8%)	737 (13.1%)	
≥80	135 (4.4%)	197 (3.5%)	
*Marital status*			<0.0001
Single	852 (27.6%)	1595 (28.4%)	
Married	2014 (65.2%)	3231 (57.5%)	
Widowed	116 (3.8%)	594 (10.6%)	
Divorced	105 (3.4%)	202 (3.6%)	
*Body mass index (kg/m²)*	25.7 ± 4.1	24.4 ± 4.9	<0.0001
*Smoking habit*			<0.0001
Never smoked	1476 (47.8%)	3182 (56.6%)	
Former smoker	602 (19.5%)	750 (13.3%)	
Current smoker	1009 (32.7%)	1690 (30.1%)	
*Occupation*			<0.0001
Class I	227 (7.4%)	456 (8.1%)	
Class II	870 (28.2%)	1156 (20.6%)	
Class III	1600 (51.8%)	1099 (19.5%)	
Class IV	390 (12.6%)	2911 (51.8%)	
*Geographic area*			0.001
Urban	1425 (46.2%)	2810 (50.0%)	
Rural	1662 (53.8%)	2812 (50.0%)	
*Previous or long-standing H. pylori infection*			0.352
No	2228 (72.2%)	4111 (73.1%)	
Yes	859 (27.8%)	1511 (26.9%)	
*Gastric cancer*			<0.0001
None	3026 (98.0%)	5583 (99.3%)	
At endoscopy	42 (1.4%)	22 (0.4%)	
Total gastrectomy	9 (0.3%)	10 (0.2%)	
Partial gastrectomy	10 (0.3%)	7 (0.1%)	

**Table 2 jcm-09-00135-t002:** Characteristics of the study participants according to the presence of gastric cancer (GC).

Variables	GC Positive	GC Negative	*p*-Value
*Total no. of patients*	64	8609	‒
*Previous or long-standing H. pylori infection*			<0.0001
No	5 (7.8%)	6300 (73.2%)	
Yes	59 (92.2%)	2308 (26.8%)	
*Thyroid disorders in males*			<0.0002
None	37 (88.1%)	2939 (97.1%)	
Hypothyroidism	4 (9.5%)	52 (1.7%)	
Hyperthyroidism	0 (0.0%)	5 (0.2%)	
Nodules in the gland	0 (0.0%)	1 (0.0%)	
Goiter	0 (0.0%)	1 (0.0%)	
Thyroidectomy	1 (2.4%)	28 (0.9%)	
*Thyroid disorders in females*			0.486
None	20 (90.9%)	4855 (87.0%)	
Hypothyroidism	1 (4.5%)	489 (8.7%)	
Hyperthyroidism	0 (0.0%)	14 (0.3%)	
Nodules in the gland	0 (0.0%)	9 (0.2%)	
Goiter	0 (0.0%)	12 (0.2%)	
Thyroidectomy	1 (4.5%)	204 (3.7%)	
*Levothyroxine therapy*			0.662
No	57 (89.1%)	7815 (90.8%)	
Yes	7 (10.9%)	794 (9.2%)	

**Table 3 jcm-09-00135-t003:** Multivariate logistic regression to estimate the risk of gastric cancer.

Variables	Males		Females	
	OR (95%CI)	*p*-Value	OR (95%CI)	*p*-Value
*Age (years)*	1.053 (1.028–1.077)	<0.0001	1.081 (1.043–1.119)	<0.0001
*Geographic area*				
Urban	Ref.		Ref.	
Rural	2.014 (1.073–3.783)	0.029	1.273 (0.572–2.836)	0.555
*Marital status*				
Single	Ref.		Ref.	
Married	2.563 (0.714–6.488)	0.091	1.518 (0.423–5.453)	0.522
Widowed	1.561 (0.061–5.750)	0.588	1.433 (0.347–5.920)	0.619
Divorced	2.394 (0.126–9.797)	0.326	‒	‒
*Smoking habit*				
Never smoked	Ref.		Ref.	
Former or current smoker	3.174 (1.492–6.755)	0.003	1.342 (0.482–3.738)	0.574
*BMI ^1^ (kg/m²)*				
<18.0	‒	‒	‒	‒
18.0–24.9	Ref.	‒	Ref.	‒
25.0–29.9	0.363 (0.188–0.701)	0.003	0.646 (0.258–1.616)	0.350
≥30.0	0.255 (0.087–0.747)	0.013	0.994 (0.319–3.098)	0.991
*Occupation*				
Class I	Ref.		Ref.	
Class II	1.277 (0.356–4.586)	0.345	1.525 (0.168–13.831)	0.707
Class III	1.149 (0.332–3.982)	0.563	1.564 (0.175–13.937)	0.689
Class IV	0.770 (0.122–4.867)	0.428	1.987 (0.247–15.637)	0.514
*Hypothyroidism*				
No	Ref.		Ref.	
Yes	5.105 (1.688–15.44)	<0.0001	0.349 (0.047–2.594)	0.303
*Previous or long‒lasting H. pylori infection*				
No	Ref.		Ref.	
Yes	2.406 (1.456–3.978)	<0.0001	1.845 (0.826–4.121)	0.135

^1^ Body Mass Index, Ref.: Category of reference for comparison.

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
