# Peer review of "Male Predominance of Gastric Cancer among Patients with Hypothyroidism from a Defined Geographic Area"

_jcm, 2020, doi:10.3390/jcm9010135_

Round 1

Reviewer 1 Report

GENERAL COMMENTS

The manuscript should be reviewed by a native English speaker.

ABSTRACT

It is stated that “The risk of cancer occurrence was not increased in the overall cohort (Odd Ratio: 1.04, n.s.)…”. However to correctly understand this statement, it should be previously explained/defined the cohort, the cases and the controls.

In the conclusion it is pointed out that “This is the first study that shows that gastric carcinogenesis induced by a hypothyroid state…”. It is too speculative to conclude, form the data of the present study, that hypothyroidism “induces” gastric carcinogenesis. The present case-control study can only suggest associations but not causality. The same comment is valid for the conclusion at the end of the Discussion section.

INTRODUCTION

The Introduction section is too long, and should be shortened.

METHODS

OK.

RESULTS

It is stated that “Interestingly, hypothyroidism was more common in patients with GC compared with patients without (7.8% versus 6.3%)”. Although this difference may be “statistically” significant, it may not be “clinically” relevant. The magnitude of the association should be better discussed in the Discussion section (for example, emphasizing that the OR in males is 5.105, although the 95% CI is quite wide: 1.688-15.44).

DISCUSSION

Please see previous comment (Abstract) about the differences between association and causality.

REFERENCES

OK.

TABLES

OK.

FIGURES

OK.

Author Response

Reviewer #1

GENERAL COMMENTS

The manuscript should be reviewed by a native English speaker.

Reply: the revised manuscript has been edited by a professional proofreading service.

ABSTRACT

2 It is stated that “The risk of cancer occurrence was not increased in the overall cohort (Odd Ratio: 1.04, n.s.)…”. However, to correctly understand this statement, it should be previously explained/defined the cohort, the cases and the controls.

Reply: Thank you for the suggestion: the abstract and conclusion were modified accordingly.

3 In the conclusion it is pointed out that “This is the first study that shows that gastric carcinogenesis induced by a hypothyroid state…”. It is too speculative to conclude, form the data of the present study, that hypothyroidism “induces” gastric carcinogenesis. The present case-control study can only suggest associations but not causality. The same comment is valid for the conclusion at the end of the Discussion section.

Reply: Thanks again, in the revised manuscript we have amended the two statements.

INTRODUCTION

4 The Introduction section is too long, and should be shortened.

Reply: We tried to satisfy your request according to the reviewer# 4, who asked for additional sentences.

RESULTS

5 It is stated that “Interestingly, hypothyroidism was more common in patients with GC compared with patients without (7.8% versus 6.3%)”. Although this difference may be “statistically” significant, it may not be “clinically” relevant. The magnitude of the association should be better discussed in the Discussion section (for example, emphasizing that the OR in males is 5.105, although the 95% CI is quite wide: 1.688-15.44).

Reply: A comment about that was added in the discussion section page 8, lines 209-212.

DISCUSSION

6 Please see previous comment (Abstract) about the differences between association and causality.

Reply: See above.

Reviewer 2 Report

This article concerns the relationship between gastric cancer (GC) and hypothyroidism in a retrospective study of   a patients group undergone an upper endoscopy for different reasons from 2002 to 2018 at the Gastroenterology Department,  University of Sassari (Italy). The aim of this paper is to demonstrate the greater presence of GC  in men (what it has already been reported in literature) and the link between GC and hypothyroidism in males compared with women evaluating many confounding variables such as Hp nfection, age, sex, smoking habit, occupation etc.

The manuscrpt though well written and including a huge amount of analyzed data, shows some flaws that I report below. The presence of Helicobacter pylori(Hp) infection, as expected, plays a crucial role in this situation resulting a determinant factor in both univariate and multivarite analyses. However data concerning previous Hp infections in my opinion are incomplete and not fully reliable. How has been Hp detected  in the endoscopy samples? Through the cultures and susceptibility testings from a microbiology laboratory? I don't think so considering that the samples refer to distant years. And why only in case of chronic active gastritis and not for other gastric pathologies where Hp might have been present,  was the infection  assessed by additional non-invasive tests? This can be an issue considering that Hp infection is determinant in this context. The greater presence of GC in men is reported in literature as well as the correlation between hypothyroidism and GC. The authors have shown that the association between GC and hypothyroidism is statistically higher in males compared with females even though in women the status of hypothyroidism due to deficiency of iodine, presence of goiter, impaired immunity etc. is more common. What is unconvincing  are the reasons  the authors give to explain this result: for example the fact that women are more prone than men  to go to the doctor so that they underwent precociously a specific therapy and then  are exposed to hypothyroidism for a shorter period or the fact that women unlike men are protected by  estrogen hormone. Anyway the limitations in this survey that the authors themselves recognize, are many: for example the limited number of subjects under study with hypothyroidism  and GC (only 5 see table 2) or the non-evaluation  of the influence of the hypothyroidism length on the onset of GC (because no follow-up has been performed)  or the absence of the diet data assessment  which is a relevant factor. Moreover I ask the authors if it is correct to say that people regardless of gender with no tyroid disorders, are more prone to contract a GC (89.1%) respect to those with tyroid problems whose values range from 0% to 7,8% (see table 2).  Lastly there are some inconsistencies to be corrected in the tables: in table 1 on first line after age decades what the numbers 59 (1.9%) and 140 (2.5%), refer to? The number related to the Hp detection  in gastric biopsies (4,454) reported on page 5 line135 does not correspond to the numbers reported in table 1 (901+1633) for Hp infection.

Author Response

Reviewer #2

1 This article concerns the relationship between gastric cancer (GC) and hypothyroidism in a retrospective study of  a patients group undergone an upper endoscopy for different reasons from 2002 to 2018 at the Gastroenterology Department,  University of Sassari (Italy). The aim of this paper is to demonstrate the greater presence of GC in men (what it has already been reported in literature) and the link between GC and hypothyroidism in males compared with women evaluating many confounding variables such as Hp infection, age, sex, smoking habit, occupation etc.

The manuscript though well written and including a huge amount of analyzed data, shows some flaws that I report below. The presence of Helicobacter pylori (Hp) infection, as expected, plays a crucial role in this situation resulting a determinant factor in both univariate and multivarite analyses. However, data concerning previous Hp infections in my opinion are incomplete and not fully reliable. How has been Hp detected in the endoscopy samples? Through the cultures and susceptibility testings from a microbiology laboratory? I don't think so considering that the samples refer to distant years.

And why only in case of chronic active gastritis and not for other gastric pathologies where Hp might have been present, was the infection assessed by additional non-invasive tests? This can be an issue considering that Hp infection is determinant in this context.

Reply: The relationship between H.  pylori and gastritis is constant and direct and invariably associated with histopathological findings of mucosal inflammation. Normal gastric mucosa is essentially devoid of inflammatory cells.  H. pylori infection leads to infiltration of the stomach with polymorphonuclear cells (active inflammation) and with mononuclear inflammatory cells (chronic inflammation) resulting in a pattern of active - chronic gastritis. The natural history of active - chronic gastritis is an extension of the inflamed area from the antrum into the corpus resulting in a reduction in acid secretion and eventually loss of parietal cells and development of atrophy. Failure to replace the loss of gastric epithelial cells by an appropriate cell proliferation may result in the replacement of an intestinal-type epithelium and eventually dysplasia, universally recognised as sequelae of a long-lasting or a past infection (Valle J, et al. Scand J Gastroenterol. 1996;31:546-50. Bayerdorffer E, et al J Clin Pathol 1989;42:834–9. Dixon MF, et al Am J Surg Pathol. 1996 Oct;20(10):1161-81, and more). This scenario occurs, especially, in infected adults with corpus inflammation and varies in different geographic regions in relation to other environmental factors. The natural history of GC includes a latent period followed by an upward inflection in which the risk increases exponentially.  This timeline reflects the extent and severity of mucosal atrophy. It has been, repeatedly, reported that H pylori eradication and endoscopy surveillance of precancerous lesions reduce significantly the risk to develop GC. For all this reasons, it is strongly recommended in clinical practice to collect histological samples in patients undergoing esophagogastroduodenoscopy (EGD). Gastric biopsies are useful not only for the diagnosis of H. pylori infection, but also provide information regarding the presence and the features of gastritis and the detection of intestinal metaplasia and or atrophy or dysplasia. The combination of 4 biopsy sites was found to be optimal for the detection of H. pylori and the assessment of the extent of atrophic gastritis (Satoh K Am J Gastroenterol. 1998; 93(4): 569). Since the number of organisms is often unequal in distribution around the stomach and are absent or very difficult to visualize in areas of intestinal metaplasia; in patients with chronic active gastritis, when H. pylori is not found in the gastric specimens the accuracy of diagnosis is improved by using one of two robust non-invasive tests such as the stool antigen test or 13C urea breath test.  Chronic gastritis is a histological entity not associated with H. pylori infection, and in this setting there is no need for additional tests to detect the bacteria.

Our endoscopy service is part of a teaching hospital devoted to research, especially in the H. pylori infection field, over the last 25 years. In clinical practice, we collect at least 4 biopsies from the antrum, angulus and body of the stomach and we placed specimens into separate bottles. According to it, we recently published a study on idiopathic peptic ulcers (H. pylori and NSAIDs negative) and the prevalence was 1.03%, one of the lowest in the literature (Dore MP, et al Scand J Gastroenterol. 2019 Nov;54(11):1315-1321)

We work according to the scientific literature in clinical practice and for each patient included in the study, we have the result about gastritis and more specifically for patients with GC, although we do not have information about previous gastritis in patient who developed GC.

2 The greater presence of GC in men is reported in literature as well as the correlation between hypothyroidism and GC. The authors have shown that the association between GC and hypothyroidism is statistically higher in males compared with females even though in women the status of hypothyroidism due to deficiency of iodine, presence of goiter, impaired immunity etc. is more common. What is unconvincing are the reasons the authors give to explain this result: for example the fact that women are more prone than men to go to the doctor so that they underwent precociously a specific therapy and then  are exposed to hypothyroidism for a shorter period or the fact that women unlike men are protected by estrogen hormone.

Reply: It is a common notion in the literature, that women on average use physician services much more often than men, leaving men at a serious lack of reporting of health problems (Marcus, A., & Siegel, J. M., 1982. Sex differences in the use of physician services: A preliminary test of the fixed role hypothesis. Journal of Health and Social Behavior, 23, 186-197). As reported by the reviewer, men exhibit a higher incidence and mortality of GC than women. It has been reported that the testosterone and estrogen levels may have a role in GC etiology, and explain male dominance in the incidence of GC In animal models it has been shown that estrogen directly promotes apoptosis in GC and reduces the size of gastric lesions caused by H. pylori. According to literature, it seemed legitimate to speculate about estrogen in the Discussion section.

3 Anyway the limitations in this survey that the authors themselves recognize, are many: for example the limited number of subjects under study with hypothyroidism and GC (only 5 see table 2) or the non-evaluation of the influence of the hypothyroidism length on the onset of GC (because no follow-up has been performed) or the absence of the diet data assessment which is a relevant factor.

Reply: Although this study has a number of limitations, essentially given by the retrospective nature of the study, nonetheless there are several strengths to take into account. First of all, the analysis was based on a large cohort of patients accurately analyzed for the presence of H. pylori infection and for hypothyroidism. Data consistency was guaranteed mainly because of the same structured questionnaire was used during the study period and all histological samples were analyzed by the same pathologist during the study period, thus minimizing discrepancies in the diagnosis. Moreover, all study participants were from Northern Sardinia, sharing the same genetic background. Although it is true that the number of GC cases was low, there were 546 patients with hypothyroidism. However, in euthyroid men the frequency of GC was 1 in 79,8, while in hypothyroid men it was 1 in 14, i.e a nearly sixfold increase, Based on the result of chi-square test  we may be wrong only one time in 2000. The difference would have been significant even with half of the GC number among the hypothyroid patients.

For the last observation, unfortunately, given the retrospective design of the study, data about diet were not collected and so not available for the multivariate analysis.

4 Moreover I ask the authors if it is correct to say that people regardless of gender with no thyroid disorders, are more prone to contract a GC (89.1%) respect to those with thyroid problems whose values range from 0% to 7,8% (see table 2).

Reply: We agree with the reviewer. In table 2 the proportion of patients with gastric cancer is only slightly higher in those with thyroid problems because the two sexes were combined. Yet the p-value is significant because the chi-square test was calculated for a 2 x 6 table in which each thyroid disorder was considered individually.

5 Lastly there are some inconsistencies to be corrected in the tables: in table 1 on first line after age decades what the numbers 59 (1.9%) and 140 (2.5%), refer to?

Reply: Maybe some problem occurs in transferring the manuscript. In Table 1, the numbers 59 and 140 refer to the number of men and women younger than 20 years old who represent 1.9% and 2.5% of the total male and female of the dataset, respectively.

6 The number related to the Hp detection  in gastric biopsies (4,454) reported on page 5 line135 does not correspond to the numbers reported in table 1 (901+1633) for Hp infection.

Reply: In the text it is specified the number of gastric specimens positive for H. pylori infection (4,454) including active-chronic gastritis and long-lasting gastritis or previous gastritis H. pylori positive. The number is correct. H. pylori chronic active gastritis is not associated with GC (for example active-chronic gastritis limited to the antrum is inversely associated with GC), instead, metaplasia and/or atrophy and/or dysplasia are considered preneoplastic lesions and they are the result of a long-lasting infection or the stigmata of a previous infection. For these reasons in the table 1 were considered the 2534 patients positive for these histological features.  

Reviewer 3 Report

The study is well performed. My only objection concerns the real satisfaction of what announced by the title, i. e. Relatuioship between gastric cancer and hypothyroidism according to Helicobacter pylori infection. What is the real correlation between H. pylori and hypothyroidism? Are they independent factors? Are sex-related ones? A less ambiguous conclusion about this point is required. 

Author Response

1 The study is well performed. My only objection concerns the real satisfaction of what announced by the title, i. e. Relationship between gastric cancer and hypothyroidism according to Helicobacter pylori infection. What is the real correlation between H. pylori and hypothyroidism? Are they independent factors? Are sex-related ones? A less ambiguous conclusion about this point is required.

Reply: We sincerely believe that the title fully reflects what we did in the study as specified in the Material and Methods section: Multivariate logistic regression models were used to estimate the risk of GC under the assumption of hypothyroidism as predictor and adjusting for potential confounders, especially for H. pylori infection, the major risk factor for gastric adenocarcinomas

Although the relationship between hypothyroidism and H. pylori infection is interesting, to explore the mutual correlation between these 2 risk factors and GC was not the purpose of our study.

However, according the reviewer suggestion a new title was provided: Male Predominance of Gastric Cancer among Patients with Hypothyroidism in a Defined Geographic Area.

Reviewer 4 Report

The work by Dore MP et al. deal with the incidence of a very common malignancy, gastric cancer, and hypothyroidism in presence or absence of Helicobacter pylori. As the same authors underlined in their discussion, this work has a big limitationcoming from the small numbers of cancers recorded [line 247].  In spite of this, this approach and the information coming from may offer a valid basis for future perspective analyses.

I have some comments/ameliorations.

In a very large cohort of 8709 participants they succeeded in evidencing a correlation between gastric adenocarcinoma occurrence, independently from H.pylori infection, in males, that they justify with an delay in replacement therapy as compared with females. Moreover, the authors hypothesize that women exposed to hypothyroidism may be less likely to develop GC than men because of a higher content of estrogens protecting them against GC. Since the authors did not quantify estrogens in their patient’s blood this part is quite speculative and should also take into consideration the very low amount of GC-affected patients in their cohort. However, the results of this work may encourage other Centers hospitalizing GC-affected patients to follow up their thyroid status, with a particular attention to males. When they wrote that ‘sex‒specific differences in food intake (e.g. less alcohol and more fruit and vegetables among women) could explain the relative protection of women from developing gastric malignancy despite treatment with levothyroxine’, it is hard to imagine to investigate this in their very small cohort of GC-affected patients, however it could be an idea for further  prospective investigation.

What about gastritis? Why didn’t they analyze gastritis occurrence in their patients ? I wonder if authors can have access to clinical data of those GC-affected patients and get information about their possible gastritis occurrence before GC development. In this way the link to test may be : untreated hypothyroidism  >gastritis>gastric cancer.

Line 43: please define younger

Line 53: Ref 17 is very old, however, since the authors have detailed the geographical positions of the other studies they should specify the geographical area also in this Ref. I wonder if hypothyroidism incidence is  the same over the world è doi: 10.1038/nrendo.2018.18.] and if there are populations (and diets) where its occurrence is higher, and if this higher incidence is similar to that one of autoimmune disorders (for instance, AAG) or gastric cancer [doi: 10.5114/pg.2018.80001.].

Please, try to introduce your subject citing papers with epidemiology similar to that one of your cohort.

Line 55: which covariates were used ?

Line 61: not so clear the difference

Line 62: also autoimmune atrophic gastritis is a risk factor for GC

Line 62: please delete ‘elegant’ that sounds more trend then scientific

Line 71: please, cite a recent  review by Krashin et al [doi: 10.3389/fendo.2019.00059. ] which summarizes the associations between thyroid hormone and cancer, with a paragraph about gastrointestinal cancer cell

Line 146: 7.8 vs. 6.3 is not so high difference

Line 148: “and/or for premalignant lesions such as metaplasia and/or dysplasia and/or atrophy” where are these data? Please, if possible, add them in table 2.

Line 149: I would resentence this ‘Treatment with levothyroxine was not more frequent in patients with GC than in patients’

Table 3: if I have well understood, were males at higher risk for GC from rural areas, never smokers (?) hypothyroidism affected and with a previous or long-lasting HP infection?

I would advice to shorten both the introduction and the discussion.

Author Response

1 The work by Dore MP et al. deal with the incidence of a very common malignancy, gastric cancer, and hypothyroidism in presence or absence of Helicobacter pylori. As the same authors underlined in their discussion, this work has a big limitation coming from the small numbers of cancers recorded [line 247].  In spite of this, this approach and the information coming from may offer a valid basis for future perspective analyses.

Reply: Thanks a lot for the appreciation of our study. We know that unfortunately (or maybe fortunately for the patients!) the number of GC cases was low, however we analysed a large size database, with an accurate diagnosis for H. pylori infection, the major risk factor for gastric carcinogenesis.

I have some comments/ameliorations.

2 In a very large cohort of 8709 participants they succeeded in evidencing a correlation between gastric adenocarcinoma occurrence, independently from H. pylori infection, in males, that they justify with an delay in replacement therapy as compared with females. Moreover, the authors hypothesize that women exposed to hypothyroidism may be less likely to develop GC than men because of a higher content of estrogens protecting them against GC. Since the authors did not quantify estrogens in their patient’s blood this part is quite speculative and should also take into consideration the very low amount of GC-affected patients in their cohort. However, the results of this work may encourage other Centers hospitalizing GC-affected patients to follow up their thyroid status, with a particular attention to males.

Reply: Several authors reported that the testosterone and estrogen levels may have a role in gastric carcinogenesis. For example, women with a longer fertility life and those on hormone replacement therapy seem to have a decreased risk of GC, and men who have been treated with estrogen for prostate cancer have a decreased risk. Use of tamoxifen in women seems to increase their risk of gastric cancer (Chandanos E, et al. Eur J Cancer. 2008 Nov;44(16):2397-403). Animal models have been shown that estrogen directly promotes apoptosis in GC and reduces the size of gastric lesions caused by H. pylori. In our study a strong male dominance of GC was observed in patients with hypothyroidism. Although, as observed by the reviewer we did not quantify estrogens in our patient’s blood, nonetheless, according to literature, it seemed legitimate to speculate about estrogen in the Discussion section.

We found very exciting the conclusion of the reviewer to encourage other Centers hospitalizing GC-affected patients to follow up their thyroid status, with a particular attention to males.

3 When they wrote that ‘sex‒specific differences in food intake (e.g. less alcohol and more fruit and vegetables among women) could explain the relative protection of women from developing gastric malignancy despite treatment with levothyroxine’, it is hard to imagine to investigate this in their very small cohort of GC-affected patients, however it could be an idea for further  prospective investigation.

Reply: thanks

4 What about gastritis? Why didn’t they analyze gastritis occurrence in their patients ? I wonder if authors can have access to clinical data of those GC-affected patients and get information about their possible gastritis occurrence before GC development. In this way the link to test may be : untreated hypothyroidism  >gastritis>gastric cancer.

Reply: The relationship between H.  pylori and gastritis is constant and direct and invariably associated with histopathological findings of mucosal inflammation. Normal gastric mucosa is essentially devoid of inflammatory cells.  H. pylori infection leads to infiltration of the stomach with polymorphonuclear cells (active inflammation) and with mononuclear inflammatory cells (chronic inflammation) resulting in a pattern of active - chronic gastritis. The natural history of active - chronic gastritis is an extension of the inflamed area from the antrum into the corpus resulting in a reduction in acid secretion and eventually loss of parietal cells and development of atrophy. Failure to replace the loss of gastric epithelial cells by an appropriate cell proliferation may result in the replacement of an intestinal-type epithelium and eventually dysplasia, universally recognised as sequelae of a long-lasting or a past infection (Valle J, et al. Scand J Gastroenterol. 1996;31:546-50. Bayerdorffer E, et al J Clin Pathol 1989;42:834–9. Dixon MF, et al Am J Surg Pathol. 1996 Oct;20(10):1161-81, and more). This scenario occurs, especially, in infected adults with corpus inflammation and varies in different geographic regions in relation to other environmental factors. The natural history of GC includes a latent period followed by an upward inflection in which the risk increases exponentially.  This timeline reflects the extent and severity of mucosal atrophy. It has been, repeatedly, reported that H pylori eradication and endoscopy surveillance of precancerous lesions reduce significantly the risk to develop GC. For all this reasons, it is strongly recommended in clinical practice to collect histological samples in patients undergoing esophagogastroduodenoscopy (EGD). Gastric biopsies are useful not only for the diagnosis of H. pylori infection, but also provide information regarding the presence and the features of gastritis and the detection of intestinal metaplasia and or atrophy or dysplasia. The combination of 4 biopsy sites was found to be optimal for the detection of H. pylori and the assessment of the extent of atrophic gastritis (Satoh K Am J Gastroenterol. 1998; 93(4): 569). Since the number of organisms is often unequal in distribution around the stomach and are absent or very difficult to visualize in areas of intestinal metaplasia; in patients with chronic active gastritis, when H. pylori is not found in the gastric specimens the accuracy of diagnosis is improved by using one of two robust non-invasive tests such as the stool antigen test or 13C urea breath test.  Chronic gastritis is a histological entity not associated with H. pylori infection, and in this setting there is no need for additional tests to detect the bacteria.

Our endoscopy service is part of a teaching hospital devoted to research, especially in the H. pylori infection field, over the last 25 years. In clinical practice, we collect at least 4 biopsies from the antrum, angulus and body of the stomach and we placed specimens into separate bottles. According to it, we recently published a study on idiopathic peptic ulcers (H. pylori and NSAIDs negative) and the prevalence was 1.03%, one of the lowest in the literature (Dore MP, et al Scand J Gastroenterol. 2019 Nov;54(11):1315-1321)

We work according to the scientific literature in clinical practice and for each patient included in the study, we have the result about gastritis and more specifically for patients with GC, although we do not have information about previous gastritis in patient who developed GC.

 The hypothesis suggested by the reviewer untreated hypothyroidism  >gastritis>gastric cancer is very interesting, although quite difficult to test.

5 Line 43: please define younger

Reply: Done: …..subjects born after the 4th decade of the twenty century

6 Line 53: Ref 17 is very old, however, since the authors have detailed the geographical positions of the other studies they should specify the geographical area also in this Ref. I wonder if hypothyroidism incidence is  the same over the world è doi: 10.1038/nrendo.2018.18.] and if there are populations (and diets) where its occurrence is higher, and if this higher incidence is similar to that one of autoimmune disorders (for instance, AAG) or gastric cancer [doi: 10.5114/pg.2018.80001.].

Reply: In the introduction, a brief sentence about the epidemiology of hypothyroidism around the world and its relation with diet, gastric cancer and autoimmune disorders was added. Page 2, lines 55-59

7 Please, try to introduce your subject citing papers with epidemiology similar to that one of your cohort.

Reply: to the best of our knowledge there are no studies similar to our cohort.

8 Line 55: which covariates were used?

Reply: Covariates are now specified, page 2, lines 61-62

9 Line 61: not so clear the difference

Reply: In the revised manuscript the reference was changed with one more representative.

10 Line 62: also autoimmune atrophic gastritis is a risk factor for GC

Reply: Correct. This was added to the risk factors

11 Line 62: please delete ‘elegant’ that sounds more trend then scientific

Reply: done.

12 Line 71: please, cite a recent  review by Krashin et al [doi: 10.3389/fendo.2019.00059. ] which summarizes the associations between thyroid hormone and cancer, with a paragraph about gastrointestinal cancer cell

Reply: Done

13 Line 146: 7.8 vs. 6.3 is not so high difference

Reply: This is true when the analysis included bot sexes. However in euthyroid men the frequency of GC was 1 in 79,8, while in hypothyroid men it was 1 in 14, i.e a nearly six fold increase, Based on the result of chi-square test  we may be wrong only one time in 2000. The difference would have been significant even with half of the GC number among the hypothyroid patients.

14 Line 148: “and/or for premalignant lesions such as metaplasia and/or dysplasia and/or atrophy” where are these data? Please, if possible, add them in table 2.

Reply: The data requested are already comprised in the table 2. In the specified in the Material and Methods section is specified that: Presence of intestinal metaplasia and/or follicular gastritis and/or dysplasia and/or atrophy in the absence of activity was considered as a stigma of a previous H. pylori infection

15 Line 149: I would resentence this ‘Treatment with levothyroxine was not more frequent in patients with GC than in patients’

Reply: Done

16 Table 3: if I have well understood, were males at higher risk for GC from rural areas, never smokers (?) hypothyroidism affected and with a previous or long-lasting HP infection?

Reply: in the table are at risk “former or current smoker”

17 I would advice to shorten both the introduction and the discussion.

Reply: Done

Round 2

Reviewer 2 Report

I appreciated the fact that the title of the manuscript has been changed, I think that the current title is  more appropriate otherwise it would have been too focused on H. pylori infection. As far as this topic is concerned there are still some issues. First of all I want to let the authors know that it was not necessary to make a presentation on the pathology of Hp (we are not in a school to learn). On the contrary they did not answer my specific question concerning the possible execution of bacterial cultures and of antibiotic testings. From which I deduce that the cultural examination has not been carried out and that the Hp diagnosis has been made only through histological tests not always so accurate as the cultures. Moreover in  case of the histology it is important the coloring used: May-Grunwald and Giemsa or Wright-Giemsa or others? Another issue regards the non-invasive tests (UBT and fecal antigen) which show different susceptibility and sensitivity. The UBT unlike the detection of fecal antigens, is considered the golden standard at this aim. Thus the authors in the chronic active gastritis where Hp was not detected, only used one technique and not both so that there could be a discrepancy between the samples tested by UBT or by antigen fecal detection. Then I suggest the authors to report  a supplementary table  where the numbers of chronic active gastritis without Hp, tested with UBT or with antigen fecal respectively, are highlighted.

As for the reply to question 3, what the authors report “in euthyroid men the frequency of GC was 1 in 79,8, while in hypothyroid men it was 1 in 14”,  must be shown in the table with the relative explanations otherwise the reader can be misled drawing wrong conclusions. The tab 2 must be formulated again. The same change may be also referred to the reply to question 4.  As far the reply to question 6 is concerned, the authors must report in the manuscript, the explication  related to the discrepancy of the numbers reported (see reply 6) for the reasons mentioned above (to mislead the readers).

This paper should be published in a Journal with high impact factor and with a very good quality of the  articles so it is mandatory to be clear, correct  and accurate in the data presentation.

Author Response

I appreciated the fact that the title of the manuscript has been changed, I think that the current title is  more appropriate otherwise it would have been too focused on H. pylori infection.

Reply: thanks.

As far as this topic is concerned there are still some issues. First of all I want to let the authors know that it was not necessary to make a presentation on the pathology of Hp (we are not in a school to learn). On the contrary they did not answer my specific question concerning the possible execution of bacterial cultures and of antibiotic testings. From which I deduce that the cultural examination has not been carried out and that the Hp diagnosis has been made only through histological tests not always so accurate as the cultures.

Reply: We sincerely regret to give the impression of a certain pedantry. We just wanted to recapitulate the background in which we felt that the reviewer's requests seemed out of place. We agree with the reviewer that the gold standard for the presence of most infectious diseases is the successful culturing of the organism. H. pylori is easily grown in an experienced laboratory but a high rate of success is dependent on experience.  In fact, there are many practical limitations that have served to make the results less than ideal.  The problems relate to the patchy location of the bacteria, the number, handling and processing of the tissue specimens, and finally the skill of the technician in culturing. For all these reasons histology is considered to give the best yield especially when gastric biopsies are obtained from predetermined locations.  The sensitivity and specificity of histology for the diagnosis of H. pylori infection are 95% and 98%, respectively. The accuracy of histologic diagnosis of H. pylori infection can be improved by using special stains such as Giemsa as we did.

Moreover in  case of the histology it is important the coloring used: May-Grunwald and Giemsa or Wright-Giemsa or others?

Reply: See above.

Another issue regards the non-invasive tests (UBT and fecal antigen) which show different susceptibility and sensitivity. The UBT unlike the detection of fecal antigens, is considered the golden standard at this aim. Thus the authors in the chronic active gastritis where Hp was not detected, only used one technique and not both so that there could be a discrepancy between the samples tested by UBT or by antigen fecal detection. Then I suggest the authors to report a supplementary table where the numbers of chronic active gastritis without Hp, tested with UBT or with antigen fecal respectively, are highlighted.

Reply: Several guidelines recommend indifferently 13C-UBT or stool antigen assay for identifying active H. pylori infection and to confirm the eradication because both tests have high sensitivity and specificity.

However, according to the request of the reviewer a supplementary table was added.

As for the reply to question 3, what the authors report “in euthyroid men the frequency of GC was 1 in 79,8, while in hypothyroid men it was 1 in 14”, must be shown in the table with the relative explanations otherwise the reader can be misled drawing wrong conclusions. The tab 2 must be formulated again.

Reply: In the revised manuscript we have modified table 2 by dividing thyroid disorders according to sex, in addition to the presence/absence of gastric cancer. As a result, the p-value has now been reported separately in the two sexes.

The same change may be also referred to the reply to question 4.  As far the reply to question 6 is concerned, the authors must report in the manuscript, the explication related to the discrepancy of the numbers reported (see reply 6) for the reasons mentioned above (to mislead the readers).

Reply: this was better explained in the revised manuscript (page 5, lines 146-149).

This paper should be published in a Journal with high impact factor and with a very good quality of the articles so it is mandatory to be clear, correct and accurate in the data presentation.

Reviewer 4 Report

The new version of the manuscript is  properly written. The authors have correctly met the different requests and exhaustively modified the text. 

Author Response

The new version of the manuscript is properly written. The authors have correctly met the different requests and exhaustively modified the text.

Reply: thanks a lot for your kind appreciation.